# Biocompatibility of Dextran-Coated 30 nm and 80 nm Sized SPIONs towards Monocytes, Dendritic Cells and Lymphocytes

**DOI:** 10.3390/nano13010014

**Published:** 2022-12-20

**Authors:** Lisa Zschiesche, Christina Janko, Bernhard Friedrich, Benjamin Frey, Julia Band, Stefan Lyer, Christoph Alexiou, Harald Unterweger

**Affiliations:** 1Department of Otorhinolaryngology, Head and Neck Surgery, Section of Experimental Oncology and Nanomedicine (SEON), Else Kröner-Fresenius-Stiftung Professorship, Universitätsklinikum Erlangen, 91054 Erlangen, Germany; 2Department of Oral and Maxillofacial Surgery, Friedrich-Alexander-Universität Erlangen-Nürnberg (FAU), 91054 Erlangen, Germany; 3Department of Radiation Oncology, Universitätsklinikum Erlangen, 91054 Erlangen, Germany; 4Department of Otorhinolaryngology, Head and Neck Surgery, Section of Experimental Oncology and Nanomedicine (SEON), Professorship for AI-Controlled Nanomaterials, Universitätsklinikum Erlangen, 91054 Erlangen, Germany

**Keywords:** superparamagnetic iron oxide nanoparticles (SPION), magnetic resonance imaging (MRI), immune response, biocompatibility, nanomedicine

## Abstract

Dextran-coated superparamagnetic iron oxide nanoparticles (SPION^Dex^) of various sizes can be used as contrast agents in magnetic resonance imaging (MRI) of different tissues, e.g., liver or atherosclerotic plaques, after intravenous injection. In previous studies, the blood compatibility and the absence of immunogenicity of SPION^Dex^ was demonstrated. The investigation of the interference of SPION^Dex^ with stimulated immune cell activation is the aim of this study. For this purpose, sterile and endotoxin-free SPION^Dex^ with different hydrodynamic sizes (30 and 80 nm) were investigated for their effect on monocytes, dendritic cells (DC) and lymphocytes in concentrations up to 200 µg/mL, which would be administered for use as an imaging agent. The cells were analyzed using flow cytometry and brightfield microscopy. We found that SPION^Dex^ were hardly taken up by THP-1 monocytes and did not reduce cell viability. In the presence of SPION^Dex^, the phagocytosis of zymosan and *E. coli* by THP-1 was dose-dependently reduced. SPION^Dex^ neither induced the maturation of DCs nor interfered with their stimulated maturation. The particles did not induce lymphocyte proliferation or interfere with lymphocyte proliferation after stimulation. Since SPION^Dex^ rapidly distribute via the blood circulation in vivo, high concentrations were only reached locally at the injection site immediately after application and only for a very limited time. Thus, SPION^Dex^ can be considered immune compatible in doses required for use as an MRI contrast agent.

## 1. Introduction

With advances in imaging technology and increasingly accurate diagnostics with more precise resolutions, diseases can be detected even faster and lesions can be assessed even better. Magnetic resonance imaging (MRI) has an important role in imaging and to significantly increase the sensitivity, a suitable, accurate and highly biocompatible contrast agent is required. Since contrast agents are applied intravenously, they have to fulfill several important prerequisites, including blood stability, no toxicity, high biocompatibility and complete elimination [1].

For years, gadolinium-based contrast agents (GBCA) have been used for MRI. Only recently, enormous side effects have been discovered due to the fact that the elimination of GBCA from the body is not as easy as initially considered. To give some examples, intravenous GBCA application is associated with nephrogenic systemic fibrosis (NSF) in patients with reduced kidney function. In addition, gadolinium residues can be found in the brain, kidneys and bones of healthy patients, especially after repeated doses [2,3,4,5,6]. As a result, the European Medicines Agency recommended the suspension of the marketing authorities for some linear gadolinium-based contrast agent in 2017 [7], underlining the urgent need for alternatives.

Superparamagnetic iron oxide nanoparticles (SPIONs) are one option for an alternative MRI contrast agent, at least for some applications, including liver, intestine, spleen, tumor, central nervous system, bone marrow, vascular or lymphatic system imaging [8,9,10,11,12]. Furthermore, SPIONs can be tuned for various physicochemical properties such as size, shape and coating, which also allows them to be used for various additional applications such as magnetic drug targeting or magnetic hyperthermia [13,14,15,16,17,18,19].

Due to their small size, SPIONs possess a large surface to volume ratio and are therefore prone to agglomeration. In order to avoid this tendency, the particles have to be stabilized by either electrostatic, steric or electrosteric means. This can be achieved by coating the SPIONs with small molecules including citric acid, maleic acid and lauric acid [20,21,22], or by coating them with a (bio-) polymer like poly(D,L-lactic-co-glycolic acid), albumin, ethyl cellulose, casein, poly(ethylene glycol) (PEG), gelatin and especially dextran [23,24,25]. During in vivo application, the particles’ surface coating is the first point of contact with the environment and thus with the immune system. As a consequence, the choice of the coating material is not only a matter of stability but also a matter of bio-, or more precisely, immunocompatibility. In this regard, dextran is a widespread particle coating material providing SPIONs with biocompatibility and particle stability.

In previous studies we investigated the influence of SPION^Dex^ on components of blood and immune cells and found that dextran-coated SPIONs (SPION^Dex^) neither harmed erythrocytes, caused complement or platelet activation, nor did they influence endothelial-monocyte interactions [26]. We also showed that SPION^Dex^ did not induce the formation of neutrophil extracellular traps (NETs) by neutrophils, the first line of immune defense. Importantly, we further showed that they did not induce complement activation-related pseudo-allergy (CARPA) reactions in pigs [26,27]. Thus, we already proved the biocompatibility of the particles in the presence of cells from the innate immune system. For the induction of adaptive immune responses, monocytes and dendritic cells are very important. Professional antigen-presenting cells (APCs) take up antigens, digest them and present peptides to lymphocytes on their major histocompatibility complex (MHC). 

Whether 30 nm and 80 nm sized dextran-coated SPIONs interfere with the functions of monocytes, dendritic cells or lymphocytes after their stimulation with pathogens or cytokines has not yet been investigated. However, as antigen-presenting or effector cells, these cells represent mandatory functions during adaptive immune responses and must be preserved, especially during clinical measures such as MRI imaging. In this study, we investigate if SPION^Dex^ interfere with monocytes, dendritic cells and lymphocytes, all immune cells relevant during adaptive immune responses.

## 2. Materials and Methods

### 2.1. Nanoparticle Synthesis and Physicochemical Characterization

Dextran-coated SPIONs (SPION^Dex^) were synthesized as previously described [26]. Briefly, Fe(II) chloride and Fe(III) chloride were mixed at a molar ratio of 1:2 in an ice-cold aqueous dextran solution and co-precipitated with ammonia under argon atmosphere. After a heat treatment of the particles at 75 °C, they were cooled to room temperature, dialyzed against water and concentrated using ultrafiltration. Crosslinking of the particles’ dextran shell was performed by addition of sodium hydroxide and epichlorohydrin. Afterwards, the particle dispersion was again dialyzed, then concentrated and subsequently sterilized via filtration through a 0.22 µm membrane. For this study, the parameters were adjusted to achieve two differently sized dextran-coated SPIONs, namely 30 nm and 80 nm, which are referred to as SPION^Dex30^ and SPION^Dex80^.

The hydrodynamic size distribution as well as the ζ potential in water at a pH of 7.4 was determined using a Zetasizer Nano ZS (Malvern Instruments, Worcestershire, UK). An MS2G magnetic susceptometer (Bartington, Witney, UK) was used to determine the particles’ magnetic volume susceptibility and the surface chemistry of the particles was investigated using an Alpha Fourier transform infrared (FTIR) device (Bruker, Billerica, MA, USA). All experiments were performed in triplicate.

### 2.2. Sterility and Endotoxin Content

For sterility testing, 100 µL of both particle types at concentrations of 500 and 100 µg Fe/mL were plated on agar plates and were then incubated for 72 h at 37 °C. Saliva in a dilution of 1:5 and 1:25 served as a positive control and water as a negative control. In addition, particles (100 and 200 µg Fe/mL) were also spiked with saliva in a dilution of 1:25 in order to analyze their potential antimicrobial features. After incubation, images were taken from the plates.

Furthermore, the endotoxin content was tested using the EndoZyme Assay (Hyglos, Bernried, Germany) according to the manufacturer’s instructions. In brief, SPION^Dex30^ and SPION^Dex80^ were mixed with the assay reagent (80 vol% assay buffer, 10 vol% substrate, 10 vol% enzyme) in a ratio of 1:1 to achieve final iron concentrations of 25 and 50 µg/mL. As controls for nanoparticle interference and to evaluate the assay performance, selected SPION samples were also spiked with 5 endotoxin units per ml (EU/mL). The samples were pipetted into black, endotoxin-free 96-well plates in duplicates, followed by their incubation at 37 °C in a SpectraMax iD3 Plate reader (Molecular Devices, San José, CA, USA) for 90 min. During incubation, samples were excited at 360 nm every 15 min and the resulting fluorescence was recorded at 465 nm. For the calculation and visualization, GraphPad was used.

### 2.3. Quantification of Iron Uptake and Viability of THP-1 Cells

The THP-1 cells were seeded in a density of 0.2 × 10^6^ cells/mL in cell culture medium containing SPION^Dex^ (50, 100 and 200 µg/mL) for 24 and 48 h. Then, SPION^Dex^ were washed several times with phosphate-buffered saline (PBS). Finally, the cell count in the cell pellets was determined using MUSE^®^ Cell Analyzer (Merck-Millipore, Billerica, MA, USA). After centrifugation, cell pellets were dissolved in 65% nitric acid for 15 min at 95 °C. After addition of 450 µL H_2_O, the iron content was determined using atomic emission spectroscopy (MP-AES, 4200 device, Agilent Technologies, Santa Clara, CA, USA). An iron solution of 1000 g/L served as an external standard (Bernd Kraft, Duisburg, Germany). Measurement was performed at a wavelength of 371,993 nm.

To analyze cell viability, 50 µL aliquots of the THP-1 cells were stained with 250 µL of Ringer’s solution (Fresenius Kabi, Bad Homburg, Germany) containing 2 µL/mL Annexin A5-fluorescein isothiocyanate (AxV-Fitc, ImmunoTools, Friesoythe, Germany) and 66.6 ng/mL propidium iodide (Sigma-Aldrich, Taufkirchen, Germany) for 20 min at 4 °C. Then, cells were measured in a Gallios flow cytometer and data analyzed using Kaluza analysis software (1.3, 2.1.) (Beckman Coulter, Brea, CA, USA). 

### 2.4. Phagocytosis of Zymosan A and E. coli by THP-1 Monocytes

A total of 0.5 × 10^6^ THP-1 cells/mL were taken up in human AB-serum (Sigma Aldrich, Taufkirchen, Germany,) and preincubated for 15 min with SPION^Dex^. Then, 5 µL of fluorescently labeled Zymosan A from *Saccharomyces cerevisiae* or 6 µL from *E. coli* (both from Sigma Aldrich) were added and incubated for 1 h. Phagocytosis was determined from analyzing the mean fluorescent index (MFI) of THP-1 in flow cytometry. 

### 2.5. Generation of Human Immature Dendritic Cells 

Monocyte-derived immature dendritic cells (iDC) were generated from human blood cells using human leucoreduction system chambers according to Schaft et al. [28]. The samples were obtained from healthy volunteers after informed consent (approved by the ethics committee of the Friedrich-Alexander-Universität Erlangen-Nürnberg, #180_13B). Briefly, peripheral blood mononuclear cells (PBMC) were isolated by density centrifugation with Lymphoflot (BIO-RAD, Feldkirchen, Deutschland) for 20 min at 850 rcf without brake. Subsequently, the PBMCs were washed 3 times with PBS containing 1 mM ethylenediaminetetraacetic acid (EDTA, Carl Roth, Karlsruhe, Deutschland) and once with RPMI. Then, the cell count was determined using a MUSE^®^ Cell Analyzer. A total of 4 × 10^6^ PBMC in 10 mL RPMI supplemented with 1% L-Glutamin, 1% 0.4 µg/mL Gentamycin (both from Gibco, UK) and 5 mL heat-inactivated 0.22 µm filtered human AB plasma (Sigma-Aldrich, Hamburg, Germany) were then seeded onto 10 cm dishes (Sarstedt, Nümbrecht, Deutschland). After incubation at 37 °C for 90 min, non-adherent cells were removed by washing 3 times with pure RPMI medium (Biochrom, Berlin, Germany). Eventually, iDCs were generated by culturing these cells for 6 days, replacing the cell medium 3 times with fresh DC medium containing 275 U/mL interleukin (IL)-4 and 800 U/mL granulocyte-macrophage colony-stimulating factor (GM-CSF, both from Miltenyi Biotec, Bergisch Gladbach, Germany). On day 6, the cells were detached from the dishes using ice-cold PBS. The cells were centrifuged at 180 rcf for 8 min, the supernatant was removed and the cells were resuspended in 10 mL of fresh DC medium.

### 2.6. Maturation of Human Dendritic Cells and Analysis by Flow Cytometry

To investigate if SPION^Dex^ inhibit DC maturation, iDCs were incubated in the presence of SPION^Dex^. Therefore, SPION^Dex^ at iron concentrations of 50, 100 and 200 µg/mL were incubated with iDCs at a cell density of 10^6^/100 µL in 6 well plates at 37 °C for 48 h. The DC medium containing IL-1β (200 U/mL), IL-6 (1000 µ/mL), tumor necrosis factor (TNF)-α (10 ng/mL, all from ImmunoTools, Friesoythe, Germany) and prostaglandin E2 (PGE2) (1 µg/mL, Pfizer, New York, USA) served as positive control, pure DC medium served as negative control. After 48 h of incubation, the matured DCs (mDCs) were harvested from the 6 well plates by washing with ice-cold PBS-EDTA. The cells were centrifuged at 200 rcf for 12 min, the supernatant removed and the cell number adjusted to 1 × 10^6^/mL with PBS containing 2 vol% fetal calf serum (FCS). To analyze the expression of surface maturation markers, 50 µL cells were stained with fluorescence labelled antibodies (CD83-PE/Cy7, CD40-PerCP/Cy5.5, CD80-APC, CD86-PerCP/Cy5.5 (all from BioLegend, San Diego, CA, USA) in PBS containing 2% FCS for 30 min. Isotype antibodies were used as controls. Subsequently, cells were washed and fixated with 100 µL PBS containing 1% paraformaldehyde. Then, cells were analyzed in flow cytometry. 

### 2.7. Lymphocyte Proliferation Assay

Peripheral blood mononuclear cells (PBMCs) were isolated from healthy donors after informed consent (approved by the ethics committee of the Friedrich-Alexander-Universität Erlangen-Nürnberg, #257_14B) and kept frozen in liquid nitrogen until further use. For the experiments, PBMCs were thawed and stained with 2.5 µM carboxyfluorescein succinimidyl ester (CFSE, Sigma Aldrich, Hamburg, Germany) in PBS for 5 min. Then, they were washed with fresh RPMI medium supplemented with 10 vol% FBS and the cell count was adjusted to 10^7^ cells per ml. 

Afterwards, 100 µL of this cell suspension was pipetted into each well of a 96 well plate. Next, the cells were treated with PBS as a negative control, with mitogen phytohaemagglutinin (PHA) (Sigma-Aldrich, Hamburg, Germany) at 2.5, 5 and 10 mg/mL as a positive control and with H_2_O as vehicle control. To analyze whether SPION^Dex^ influence lymphocyte proliferation, PBMCs were incubated with SPION^Dex^ (50, 100 and 200 µg/mL) for 72 h and then treated with either 10 µL H_2_O or PHA to induce proliferation. After further incubation for 72 h, pictures were taken with a ZEISS Axiovert 40 microscope at a 5-fold magnification. For flow cytometry analysis (Gallios, Beckman Coulter, Fullerton, CA, USA), 30 µL of each well were stained with 150 µL Ringer’s solution containing 2 µL/mL Annexin AxV (APC ImmunoTools, Friesoythe, Germany). 

### 2.8. Statistical Analysis

Data were processed in MS Excel (Microsoft, Redmont, WA, USA). For statistical analysis and creation of graphs, GraphPad PRISM 8.3.0 from GraphPad Software, Inc., (San Diego, CA, USA) was used. Data are expressed as mean ± standard deviation. For samples with normal distribution, a parametric test was used. For samples without normal distribution, a nonparametric test was used.

## 3. Results

### 3.1. Physicochemical Characterization

As shown in Figure 1A and summarized in in Table 1, the different set of parameters used in the synthesis resulted in a hydrodynamic size of 32 ± 1 nm with a polydispersity index (PDI) of 0.129 ± 0.004 for SPION^Dex30^ and 77 ± 2 nm with a PDI of 0.137 ± 0.017 for SPION^Dex80^, respectively. For both sizes, the size distribution is quite narrow. The weak surface charge of the dextran shell at pH 7 in an aqueous dispersion is reflected in the low mean ζ potential of both samples, with −4.9 ± 0.2 mV for the small and −3.7 ± 0.3 for the larger particles (see Figure 1B). In a previous study, TEM images of similar dextran-coated particles showed that the iron oxide nuclei are aligned in a branched, chain-like structure that is more pronounced the larger the particles [26]. Nevertheless, the FTIR spectra of both particles (Figure 1C) reveal the same typical peaks for dextran in the presence of iron oxide. For example, both samples possess a peak at approx. 600 cm^−1^, corresponding to the Fe-O vibrational mode of iron oxide [29]. Furthermore, dextran causes, e.g., intense C-O-C vibrational modes in a range between 1000 and 1310 cm^−1^ and CH_2_ and C-OH deformations at 1400 cm^−1^ [30]. The magnetic volume susceptibility at 1 kHz is 8.0 × 10^−4^ ± 2.4 × 10^−6^ and 12.4 × 10^−4^ ± 1.5 × 10^−6^ (both in SI units) for SPION^Dex30^ and SPION^Dex80^, respectively. Further descriptions of the particle system, including more detailed characterization of the coating, the magnetic properties as well as imaging properties can be found in our previous publications [26,27].

### 3.2. Sterility and Endotoxin Content

To prove the sterility of SPION^Dex^, the nanoparticles were plated on agar and incubated for 72 h. Sterile H_2_O or diluted saliva served as the negative and positive control, respectively. In order to take potential particle-induced bacterial proliferation inhibition into account, the particles were also incubated in the presence of saliva. As depicted in Figure 2A, neither in the control with water nor in the agar plates treated only with SPION^Dex^ were any bacterial colonies detected regardless of their size and tested concentration, while the agar plates treated with saliva at different dilutions showed bacterial colonies. In addition, the presence of the particles at the given concentrations seemed to have neither induced, nor reduced the growth of bacteria on agar plates by visual examination (Figure 2B).

To evaluate the endotoxin level of the nanoparticles, the EndoZyme recombinant factor C assay was used [31]. As shown in Figure 2C, the detected endotoxin amounts were 0.010 EU/mL for SPION^Dex30^ and 0.015 EU/mL for SPION^Dex80^ (both at an iron concentration of 25 µg/mL). In order to exclude potential interference by the dark-colored nanoparticles during fluorescence measurements, the samples were also spiked with 5.0 EU/mL lipopolysaccharides. The recovery for SPION^Dex30^ and SPION^Dex80^ was 97% and 95%, respectively, indicating correct assay performance in the tested concentrations.

### 3.3. Uptake of SPION^Dex^ by Monocytes and Influence on Viability and Phagocytosis

As phagocytic cells of the innate immune system, monocytes circulate in blood and after migration to tissues they can differentiate into macrophages or dendritic cells. Monocytes inactivate pathogens by phagocytosis and degranulation. After the procession of engulfed material, they can present peptides on their MHC receptors, thus having a central role also for the adaptive immune system. Here we used the human THP-1 cell line as a model for human monocytes to investigate the influence of SPION^Dex^. 

First, we incubated SPION^Dex^ with monocytes for 24 h and analyzed if the nanoparticles were taken up by the cells. As a positive control, we used citrate-coated SPIONs (SPION^Cit^), which we have previously used to load T cells or fibroblasts to enable their magnetic targeting [32,33]. Measurements in AES show that in contrast to SPION^Cit^, neither SPION^Dex30^ nor SPION^Dex 80^ were taken up significantly by the monocytes (Figure 3A). This finding was confirmed in flow cytometry, where the increase of the side scatter served as a measure for nanoparticle uptake. Again, the uptake by monocytes was detected only for SPION^Cit^ (Figure 3B). Even after 48 h of incubation, all nanoparticles were biocompatible and did not reduce cell viability as determined by staining with AxV and PI (Figure 3C). Next, we wanted to investigate if SPION^Dex^ interfered with the phagocytosis of pathogens. We preincubated THP-1 with SPION^Dex^ and used fluorescent-labelled Zymosan A and *E. coli*, which where opsonized using AB-plasma. The THP-1 cells were then incubated with the model pathogens and after 1 h, the mean fluorescence index (MFI) of the THP-1 cells was determined. We found that the presence of SPION^Dex^ dose-dependently reduces the phagocytosis of Zymosan. The higher the SPION^Dex^ concentration, the lower the phagocytosis rate (Figure 3D). For SPION^Dex30^, a dose-dependent effect was observable also with *E. coli*. The SPION^Dex 80^ slightly inhibited phagocytosis of *E. coli*, but was similar in all concentrations and not significant (Figure 3E). 

### 3.4. Maturation of Dendritic Cells (DCs) in the Presence of SPIONs 

Next, we investigated the influence of SPION^Dex^ on the maturation of DCs. We wanted to know if SPION^Dex^ themselves induced DC maturation or if SPION^Dex^ interfered with stimulated maturation. Thus, we incubated immature DCs with SPION^Dex^ in the absence or presence of a stimulation mix containing IL-1β, IL-6, PGE and TNF-α. After 48 h, brightfield microscopy was used to examine the morphology of DCs. Untreated DCs were adherent and showed the typical dendritic extension. This morphology was also observed in the SPION^Dex^-treated cells, independent of size and tested concentrations. The DCs incubated with inflammatory cytokines lacked dendritic extensions and lost their adherence (Figure 4A). The cells were round-shaped and easy to resuspend. Incubation with SPION^Dex^ in the presence of the stimulation did not prohibit the maturation of the DCs as reflected by their morphology (Figure 4B).

The experiment was performed with isolated DCs from three donors. The data from the donors with the strongest response for CD80, CD83 and CD86 are presented (Figure 4C), as well as the merged normalized data for CD86 from all three donors (Figure 4D). In the presence of SPION^Dex^ alone, no maturation of the DCs was induced as shown by the lack of upregulation of the activation markers. As soon as the inflammatory cytokines were present, the activation markers CD80, CD83 and CD86 were upregulated, independently of whether SPION^Dex^ were added or not. Thus, we conclude that SPION^Dex^ itself does not induce neither inhibits the cytokine induced maturation. 

### 3.5. Lymphocyte Proliferation

Lymphocytes contribute to humoral and cellular immune responses through the recognition of antigens and production of antibodies (B cells), or the direct killing of suspicious cells (cytotoxic T cell or NK cell) or support in destruction of them (T helper cell). Thus, the lymphocytes represent central cells in the immune system. We analyzed if SPION^Dex^ induced or suppressed the induced proliferation of lymphocytes. For that, we isolated PBMCs from whole blood and stained the cells with CFSE and incubated them with SPION^Dex30^ or SPION^Dex80^. Lymphocyte proliferation was stimulated with the mitogen PHA.

When we analyzed the cells after 72 h using microscopy, we detected proliferation clusters in the PHA-stimulated cells, while no proliferation clusters were detected in the presence of SPION^Dex^ alone (Figure 5A). Contrarily, when we co-incubated the lymphocytes with both PHA and SPION^Dex^, we detected proliferation clusters also in the presence of SPION^Dex^, indicating that the nanoparticles did not suppress the proliferation of cells (Figure 5B). These observations achieved in transmission microscopy were confirmed in flow cytometry, where we counted the cells and analyzed the CFSE fluorescence of the cells. Non-stimulated lymphocytes showed around 1000 cells per measurement while with PHA the cell count increased to 2500 cells, independently of whether SPIONs were present or not (Figure 5C). Due to cell proliferation, the intracellular CFSE fluorescence was distributed to the daughter cells, as depicted after stimulation with PHA. Again, SPION^Dex^ alone did not decrease CFSE fluorescence, while all PHA-treated samples showed a reduced CFSE fluorescence, independently of whether SPION^Dex^ were present or not (Figure 5D).

## 4. Discussion

Dextran-coated superparamagnetic iron oxide nanoparticles are intended for use as MRI CA; therefore, they must be toxicologically safe, regardless of the used size. That includes biocompatibility towards various cells of the blood system and soluble plasma components, which we confirmed previously [26,27]. The rationale behind the two chosen sizes of 30 and 80 nm lies in the size-dependent distribution behavior. While smaller particles in the range between 20 and 40 nm are able to visualize atherosclerotic plaques due to a longer blood circulation time [34], larger particles accumulate more preferentially in the liver and lymph nodes [26]. In our current study, we investigated if SPION^Dex^ interferes with the activation of immune cells, which can become important if there is an infection or other insult during the application of SPION^Dex^ contrast agent. 

When SPION^Dex^ are applied to the bloodstream, they reach high concentrations for a limited time locally. Thus, immune cells circulating in the blood such as monocytes, immature dendritic cells and lymphocytes will get in contact with the particles. After uptake of a pathogen in dendritic cells, intracellular degradation and processing, peptides are presented on the major histocompatibility complex (MHC) to lymphocytes [35]. Dendritic cells undergo maturation characterized by morphological changes and expression of surface markers and cytokines [36]. Interestingly, the various types of nanoparticles have been reported to be taken up and to influence the process of dendritic cell maturation and by that, the immune response. Exemplarily, C60 fullerenes and carbon black induced dendritic cell maturation, while 10 nm-sized gold nanoparticles inhibited the LPS-induced expression of CD86, CD83 and IL-12p70 103 [37,38,39,40]. When we incubated SPION^Dex^ with immature dendritic cells, we did not detect the induction of maturation by the particles (Figure 4); neither, did we observe the interference of SPION^Dex^ with maturation induced by a mix of inflammatory cytokines (Figure 4). Previously, SPIONs were used to label dendritic cells to follow their in vivo migration using MRT, receiving a good cellular uptake [41,42]. For the investigation of the uptake of SPION^Dex^, we used THP-1 monocytes, where we hardly detected any uptake of the particles, while citrate-coated particles were taken up well (Figure 3A,B). This is in line with earlier findings of weak cellular uptake of SPION^Dex^ by cells analyzed in our group [43]. The activation of monocytes has been ascribed to iron oxide nanoparticles, leading to the production of secondary messengers and then endothelial dysfunction and finally atherosclerosis. However, it seems that rather the coating and/or other physicochemical characteristics are decisive for the interactions with monocytes because particles coated with starch, PVA or PEG induced inflammatory monocyte reactions, while dextran-coated particles did not [44,45,46]. When *E. coli* or zymosan were added to the THP-1 cells in SPION^Dex^-containing medium, uptake of both was dose-dependently reduced by SPION^Dex^ (Figure 3D,E). We assume that SPION^Dex^ interfere with pathogen recognition by phagocytic receptors. In line with our findings, a decrease in phagocytosis of *E. coli* by RAW macrophages was described for 10 nm gold nanoparticles and 10 nm SiO_2_ nanoparticles, which, however, had been taken up by the cells [47].

In the blood or peripheral organs, immature DCs take up antigens before migrating to the draining lymph nodes where they undergo maturation and interact with lymphocytes. When we tested the interaction of SPION^Dex^ with lymphocytes (T and B cells), we neither detected any induction of proliferation by the particles nor inhibition of proliferation induced by PHA (Figure 5). Previously, iron oxide nanoparticles were also used to label T cells for MRI. Therefore, several investigations exist, which are, however, inconsistent [48]. Beside the detection of any effects of SPION labelling of the cells, a delay in proliferation rates and cytotoxicity have also been detected [49,50]. Again, these findings coincide with our data on the loading of T cells intended for the magnetic targeting. Whether the particles showed toxic effects was dependent on the physiochemical characteristics and the coating, not the iron cores [33,51,52]. When incubating with PMN, the first cell defense line in blood, we saw the same effect: while coating with lauric acid did not mask the SPIONs from causing NETosis, coating with dextran passivated the SPIONs so that no reaction from PMN was caused, even when accumulated with the magnetic fields [53]. For all of the reactions, the presence of plasma proteins around the particles might also play a role, forming a protein corona around the particles and preventing them from cellular uptake and causing inflammatory reactions [44]. 

In sum, we showed that SPION^Dex^ neither introduced nor interfered with stimulated DC maturation or lymphocyte proliferation. Only in high concentrations, the suppressive effects on phagocytosis by monocytes were observed. These results suggest that SPION^Dex^ are highly biocompatible and do not interfere with the immune system in dosages relevant for MRI. These results are consistent with those presented by Unterweger et al. [27], which showed that SPION^Dex^ does not cause complement, platelet activation, CARPA reactions or coagulations, affected leukocyte PCA, or influence unstimulated leukocyte proliferation and endothelial-monocytic cell interactions.

## Figures and Tables

**Figure 1 nanomaterials-13-00014-f001:**
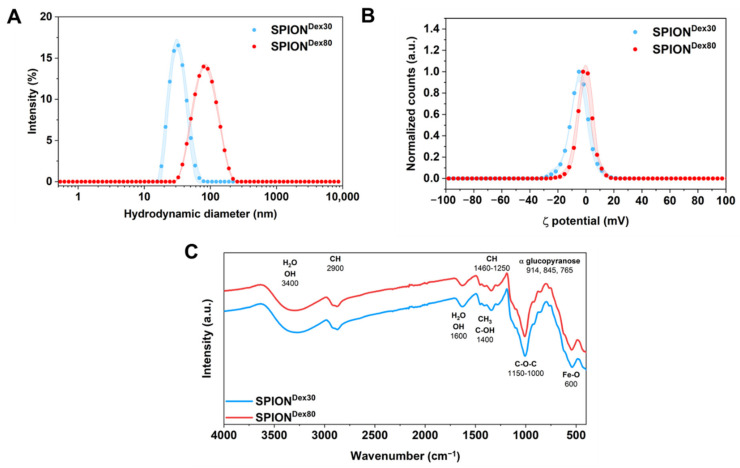
Physicochemical properties of differently sized SPION^Dex^. (**A**) Intensity-weighted hydrodynamic size distribution of the particles with shadows showing the standard deviations of three independent experiments. (**B**) Normalized ζ potential distribution with shadows showing the standard deviations of three independent experiments. (**C**) FTIR spectra of SPION^Dex30^ and SPION^Dex80^ with peak identification according to [30] reveal that there is no difference in surface chemistry.

**Figure 2 nanomaterials-13-00014-f002:**
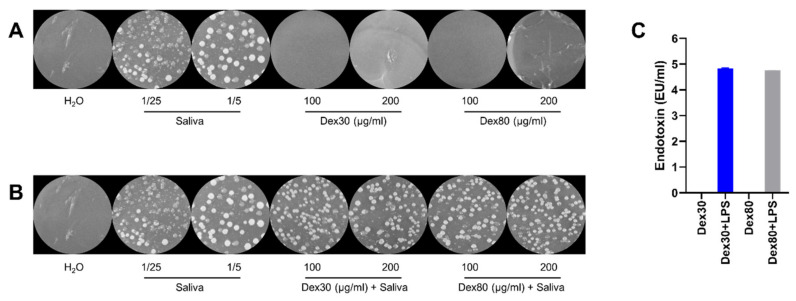
SPION^Dex^ are free of bacterial and endotoxin contaminations. (**A**) Agar plates were incubated with differently sized dextran-coated SPIONs for 72 h. H_2_O was used as negative and water-diluted saliva (1:5 and 1:25) as positive controls. (**B**) SPIONs supplemented with 1:25 diluted saliva neither inhibited nor enhanced bacterial growth. (**C**) According to the EndoZyme assay, both particle types have an endotoxin level of <0.2 EU/mL at an iron concentration of 25 µg/mL. SPION^Dex^ spiked with 5 EU/mL lipopolysaccharides served as the interference control group. The experiment was performed in duplicates. The mean values with standard deviation are shown.

**Figure 3 nanomaterials-13-00014-f003:**
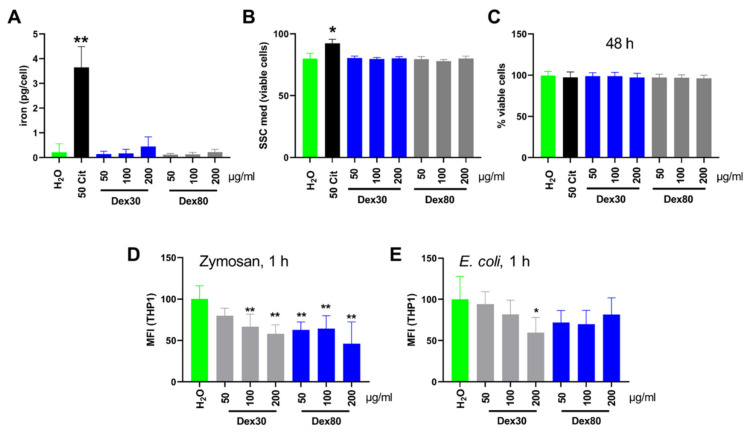
The effect of 30 nm and 80 nm SPION^Dex^ on the THP-1 cells. The THP-1 cells were incubated with 50, 100 and 200 µg/mL SPION^Dex30^ and SPION^Dex80^ or SPION^Cit^, serving as a positive control. The nanoparticle uptake by the cells was determined by atomic emission spectroscopy (**A**) or side scatter in flow cytometry (**B**). The viability of the THP-1 cells was determined after 48 h of nanoparticles incubation by AxV and PI staining in flow cytometry. The AxV-PI-cells can be regarded as viable. The amount of viable cells in the negative control (H_2_O) was set to 100% and viability calculated accordingly (**C**). (**D**,**E**) Opsonized fluorescent-labelled Zymosan A and *E. coli* were added to THP-1 preincubated with SPION^Dex^. Phagocytosis by the THP-1 cells was estimated using their mean fluorescence index. The experiment was performed in three (**A**–**C**) or two (**D**,**E**) independent triplicates. The mean values with standard deviations are shown. Significance was analyzed using a Kruskal–Wallis test (** *p* ≤ 0.01, * *p* ≤ 0.05).

**Figure 4 nanomaterials-13-00014-f004:**
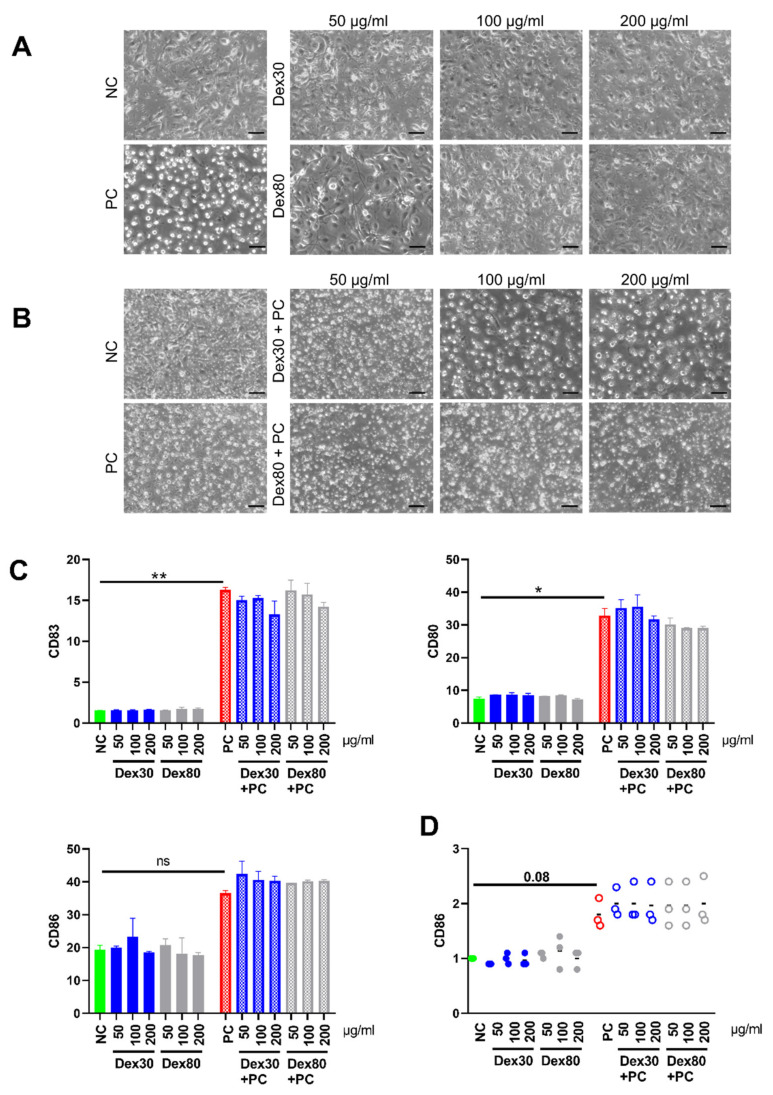
The effect of 30 nm and 80 nm SPION^Dex^ on the maturation of DCs. Immature DCS were incubated with SPION^Dex^ in the absence or presence of a stimulation mix containing IL-1β, IL-6, PGE and TNF-α (as positive control, PC) for 48 h. The experiment was performed with blood from three healthy donors and PBS served as negative control (NC). (**A**,**B**) Brightfield microscopy of (**A**) dendritic cells after 48 h incubation with SPION^Dex^, (**B**) dendritic cells after 48 h incubation with SPION^Dex^ and stimulation mix. Scale bar refers to 200 µm. (**C**) Flow cytometry of DC maturation markers CD80, CD83, and CD86 on DCS from one exemplary donor. (**D**) Merged data of the three donors. The negative control of each donor was set to 100, samples were calculated accordingly. The mean values for each donor are shown. Black line represents the mean values of all donors. Significance was calculated using a Kruskal–Wallis test (** *p* ≤ 0.01, * *p* ≤ 0.05).

**Figure 5 nanomaterials-13-00014-f005:**
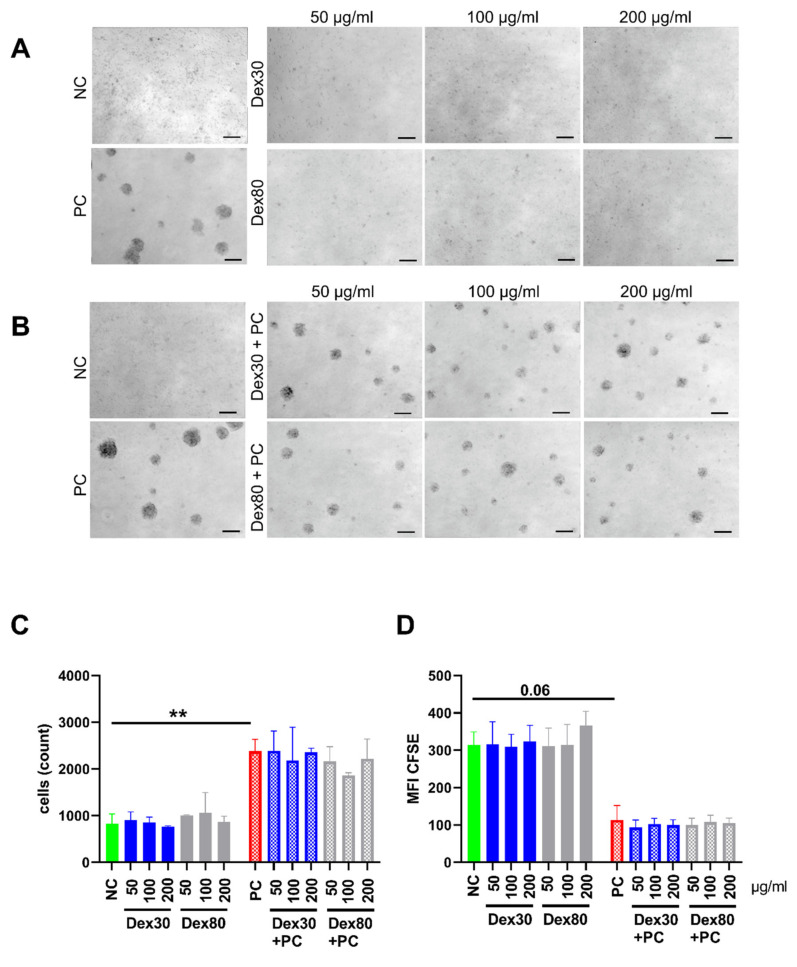
Effect of 30 nm and 80 nm SPION^Dex^ on lymphocyte proliferation. Lymphocytes were incubated for 72 h with SPION^Dex^ in the absence or presence of PHA (as the positive control, PC), activating lymphocytes. The experiment was performed with blood from three healthy donors and PBS served as a negative control (NC). Scale bar refers to 400 µm. (**A**,**B**) Brightfield microscopy of (**A**) lymphocytes after 72 h incubation with SPION^Dex^, or (**B**) lymphocytes after 72 incubation with SPION^Dex^ and PHA. (**C**) Flow cytometric analysis of viable cell count, (**D**) mean fluorescence index (MFI) of carboxyfluorescein succinimidyl ester (CFSE) fluorescence. The experiment was performed with blood from three donors in triplicates. The mean values with standard deviations are shown. Significance was calculated using a Kruskal–Wallis test (** *p* ≤ 0.01).

**Table 1 nanomaterials-13-00014-t001:** Comparison of basic physicochemical parameters of SPION^Dex^ with a mean size of 30 and 80 nm, respectively.

Parameter	SPION^Dex30^	SPION^Dex80^
Hydrodynamic size (nm)	32 ± 1	77 ± 2
PDI (a.u.)	0.129 ± 0.004	0.137 ± 0.017
ζ potential (mV) at pH 7	−4.9 ± 0.2	−3.7 ± 0.3
Magnetic volume susceptibility (SI units)	8.0 × 10^−4^ ± 2.4 × 10^−6^	12.4 × 10^−4^ ± 1.5 × 10^−6^

## Data Availability

The data presented in this study are available on request from the corresponding author.

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
