# Peer review of "Biocompatibility of Dextran-Coated 30 nm and 80 nm Sized SPIONs towards Monocytes, Dendritic Cells and Lymphocytes"

_nanomaterials, 2022, doi:10.3390/nano13010014_

Round 1

Reviewer 1 Report

The manuscript presented by Lisa Zschiesche and collaborators describes the effects of dextran-coated superparamagnetic nanoparticles (SPIONDex) on cells of the immune system after their activation with pathogens or cytokines.

In the first part of the study, the physicochemical properties of nanoparticles of different sizes, namely their zeta potential and spectroscopic characteristics were analyzed by FTIR.

Next, the authors studied the effect of SPIONDex on monocytes, dendritic cells and lymphocytes at concentrations up to 200 μg/ml using flow cytometry and bright-field microscopy. They found that SPIONDex were hardly taken up by THP-1 monocytes and did not reduce cell viability. In the presence of SPIONDex, phagocytosis of zymosan and E-coli by THP-1 was reduced in a dose-dependent manner. SPIONDex neither induced the maturation of DCs nor interfered with their stimulated maturation. The particles neither induced lymphocyte proliferation nor interfered with lymphocyte proliferation after stimulation.

The manuscript is clearly written and rich in experimental data. The results are well presented and the data clearly support the conclusions. The research is of interest to readers of the journal and contains a novel element (i.e., the lack of interference of SPIONDex during adaptive  immune response).

The authors could improve the manuscript by addressing the following points:

1) The authors studied SPIONDex of two different sizes. The choice of size is certainly supported by previous investigations. The authors should briefly discuss the reason for choosing this size range, especially n relation to the final application of the nanoparticles.

2) In a previous work, the authors have shown that SPIONDex are also poorly internalized in endothelial and other cell lines. If SPIONDexes do not accumulate in tissues or organs, for what kind of application could they be useful? This point should be clearly explained for clarity.

3) The optical images are very small and only large field of views are provided. It is hard to appreciate details of the cells.

Minor point:

-       Specify in the captions of Figs. 4 and 5 the abbreviations PC and NC for the two datasets.

Also check the text for some missing definitions of the abbreviations used.

Author Response

Der Reviewer, 

Thank you for the evaluation of our manuscript. In the following we would like to address your points in more detail

  • The authors studied SPIONDex of two different sizes. The choice of size is certainly supported by previous investigations. The authors should briefly discuss the reason for choosing this size range, especially n relation to the final application of the nanoparticles.

Thank you, that is indeed a critical point that we missed to clarify in the original manuscript. We added the following part to the beginning oft he discussion:

„SPIONDex are intended for use as MRI CA, therefore they must be toxicologically safe, regardless of the used size. That includes biocompatibility towards various cells of the blood system and soluble plasma components, which we confirmed previously [26, 27]. The rationale behind the two chosen sizes of 30 and 80 nm lies in the size dependent distribution behavior. While smaller particles in the range between 20 and 40 nm are able to visualize atherosclerotic plaques due to a longer blood circulation time [34], larger particles accumulate more preferentially in the liver and lymph nodes [26].“

  • In a previous work, the authors have shown that SPIONDex are also poorly internalized in endothelial and other cell lines. If SPIONDexes do not accumulate in tissues or organs, for what kind of application could they be useful? This point should be clearly explained for clarity.

While it is true that SPIONDex is poorly internalized in many different cell types, we have previously shown that they can be internalized by macrophages an they can, depending on their size, accumulate in different regions of the body, for example atherosclerotic plaques, liver, lymph nodes, etc. That is why we can use these particles as MRI contrast agents for these regions (see question 1).

  • The optical images are very small and only large field of views are provided. It is hard to appreciate details of the cells.

Optical images (in figure 4 and 5) were revised in order to reduce the field of view and to improve the visibility of cell details.

  • Minor point: Specify in the captions of Figs. 4 and 5 the abbreviations PC and NC for the two datasets. Also check the text for some missing definitions of the abbreviations used.

Thank you. The mentioned abbreviations were specified in figure 4 and 5. Furthermore, we went through the whole manuscript to include missing definitions.  

Reviewer 2 Report

Authors tested interaction of the dextran-coated superparamagnetic iron oxide nanoparticles (SPIOND) of various sizes on monocytes, dendritic cells and lymphocytes, all immune cells relevant during adaptive immune responses.  Authors showed that SPIOND neither introduce nor interfere with stimulated DC maturation or lymphocyte proliferation. Only in high concentrations suppressive effects on phagocytosis by monocytes were observed. So, SPIOND are highly biocompatible and do not interfere with the immune system in dosages relevant for MRI. 

More emphasis should be placed on a detailed description of the properties of the material that covers the nanoparticles and on the combination of both materials (nanoparticle and coating). Authors should show not only interaction of SPIOND with selected cells, but also how they behave in a magnetic field (as a complex for MRI).

Author Response

Dear Reviewer, 

Thank your for your evaluation of our manuscript.  We adresser your points as follows: 

  • More emphasis should be placed on a detailed description of the properties of the material that covers the nanoparticles and on the combination of both materials (nanoparticle and coating). Authors should show not only interaction of SPIOND with selected cells, but also how they behave in a magnetic field (as a complex for MRI).

Thank you for your review. We already have a detailed description of our particles including the rationale about the coating in our previous publications, which are referenced on a few occasions in the outlined manuscript. The same is true for more detailed magnetic properties (including saturation magnetization, relaxivity values, etc). We included the following sentence to the end of the physicochemical section to make it more clear:

„Further descriptions of the particle system, including more detailed characterization of the coating, the magnetic properties as well as imaging properties can be found in our previous publications [26, 27].“

  1. Unterweger, H., et al., Non-immunogenic dextran-coated superparamagnetic iron oxide nanoparticles: a biocompatible, size-tunable contrast agent for magnetic resonance imaging. International Journal of Nanomedicine, 2017. 12: p. 5223-5238.
  2. Unterweger, H., et al., Dextran-coated superparamagnetic iron oxide nanoparticles for magnetic resonance imaging: evaluation of size-dependent imaging properties, storage stability and safety. International Journal of Nanomedicine, 2018. 13: p. 1899-1915.

Round 2

Reviewer 2 Report

Can be published in the present form.